# The Mitotic Exit Network integrates temporal and spatial signals by distributing regulation across multiple components

Ian Winsten Campbell, Xiaoxue Zhou, Angelika Amon*

David H. Koch Institute for Integrative Cancer Research, Howard Hughes Medical Institute, Massachusetts Institute of Technology, Cambridge, United States

**Abstract** GTPase signal transduction pathways control cellular decision making by integrating multiple cellular events into a single signal. The Mitotic Exit Network (MEN), a Ras-like GTPase signaling pathway, integrates spatial and temporal cues to ensure that cytokinesis only occurs after the genome has partitioned between mother and daughter cells during anaphase. Here we show that signal integration does not occur at a single step of the pathway. Rather, sequential components of the pathway are controlled in series by different signals. The spatial signal, nuclear position, regulates the MEN GTPase Tem1. The temporal signal, commencement of anaphase, is mediated by mitotic cyclin-dependent kinase (CDK) phosphorylation of the GTPase's downstream kinases. We propose that integrating multiple signals through sequential steps in the GTPase pathway represents a generalizable principle in GTPase signaling and explains why intracellular signal transmission is a multi-step process. Serial signal integration rather than signal amplification makes multi-step signal transduction necessary.
DOI: https://doi.org/10.7554/eLife.41139.001

*For correspondence:
angelika@mit.edu

Competing interests: The authors declare that no competing interests exist.

## Introduction

Accurate chromosome segregation is essential for genome maintenance and viability of the organism. Central to this accuracy is the integration of temporal and spatial signals by the cell cycle machinery. Deciphering the molecular mechanisms that bring about signal integration is thus critical to understanding how organisms are built.

In eukaryotes, a family of protein kinases, known as cyclin-dependent kinases (CDKs) drive progression through the cell cycle. G1 CDKs promote cell cycle entry, S phase CDKs initiate DNA replication, and mitotic CDKs drive entry into mitosis (reviewed in *Morgan, 1997*). Events during mitosis are controlled by a ubiquitin ligase known as the Anaphase Promoting Complex or Cyclosome (APC/C; *Figure 1A*; reviewed in *Peters, 2006*; *Thornton and Toczyski, 2006*). The APC/C together with its specificity factor Cdc20 degrade the anaphase inhibitor Securin (Pds1 in budding yeast; *Shirayama et al., 1999*), to allow the protease Separase (Esp1 in budding yeast) to cleave cohesins, the proteins that hold the duplicated sister chromatids together. Cleavage of cohesins initiates anaphase. APC/C-Cdc20 not only targets Securin for degradation, the ubiquitin ligase also ubiquitylates S phase and mitotic cyclins (*Irniger et al., 1995*; *King et al., 1995*; *Sudakin et al., 1995*; *Tugendreich et al., 1995*). In budding yeast, APC/C-Cdc20 promotes the complete destruction of the S phase cyclin Clb5 and degradation of approximately half the pool of the mitotic cyclin Clb2 (*Shirayama et al., 1999*). This partial degradation of cyclins is a prerequisite for execution of the final cell cycle transition – exit from mitosis.

**Figure 1.** MEN activation requires APC/C activity. (**A**) Control of the metaphase – anaphase transition. APC/C^Cdc20 has three roles in activating the MEN: cyclin degradation (1), activating the FEAR network (2), and triggering chromosome segregation (3). See text for more details. (**B**) The zone model explains spatial regulation of the MEN. Anaphase spindle elongation along the mother-bud axis drives one Tem1 bearing SPB into the bud. This removes the MEN GTPase from the inhibitory effect of Kin4 and brings it into contact with the MEN activator Lte1, activating the MEN. If spindle elongation fails to translocate a SPB into the bud the MEN does not become active. During metaphase SPBs can make excursions into the bud. We assayed MEN activity during these pre-anaphase excursions. (**C, D, E**) *Wild type* (A39323), *cdc23-1* (A39374), *cdc23-1 mcd1-1* (A40568), and *cdc23-1 mcd1-1 mad1Δ* (A39461) cells containing *MOB1-eGFP*, *SPC42-mCherry*, and *CDC14-tdTomato* were grown at 34°C and imaged every 3 min for 4–6 hr. Curves show mean and stdev. (n = 20 cells per condition; see *Figure 1—figure supplement 2* for individual traces and sample images). Data were normalized to the average intensity measured during the 15 min prior to anaphase. Data were centered at the frame where at least one SPB had moved into the daughter for the first time. This centered timepoint was designated t = 3 min. (**C**) Mob1-eGFP intensity at mother and daughter cell SPBs [max(mSPB)+max(dSPB)]. (**D**) Cdc14 release from the nucleolus measured as coefficient of variation (CV = stdev/mean) of Cdc14 intensity within the cell. (**E**) Percent cells that exited mitosis and entered a new cell cycle as judged by their ability to form a new bud (rebudding). (**F**) *MET-CDC20 mad1Δ* (A37828), *MET-CDC20 mad1Δ mcd1-1* (A37907), and *mad1Δ mcd1-1* (A37739) cells containing *NUD1-3V5* were arrested in G1 with α-factor (5 μg/ml) pheromone at room temperature in synthetic medium lacking methionine. After 3 hours cells were released into pheromone-free YEPD medium supplemented with 8 mM methionine at 37°C. Methionine was re-added every hour. The percentage of cells with buds (black), metaphase spindles (blue), and anaphase spindles (red) was determined at the indicated times. Nud1 T78 phosphorylation and total Nud1 levels were determined.

DOI: https://doi.org/10.7554/eLife.41139.002

*Figure 1 continued on next page*

*Figure 1 continued*

The following figure supplements are available for figure 1:

**Figure supplement 1.** *CDC15* mutants do not exhibit Kar9 localization and SPB inheritance defects.

DOI: https://doi.org/10.7554/eLife.41139.003

**Figure supplement 2.** Mob1 localization to SPBs and release of Cdc14 from the nucleolus depends on the APC/C.

DOI: https://doi.org/10.7554/eLife.41139.004

During exit from mitosis mitotic events are reversed. The mitotic spindle is disassembled, chromosomes decondense and cytokinesis ensues. Mitotic CDK inactivation brings about this transition (*Bardin and Amon, 2001*; *Weiss, 2012*). In mammals, CDK inactivation occurs in a single step initiated at the metaphase to anaphase transition. In budding yeast CDK inactivation is a two-step process (reviewed in *Sullivan and Morgan, 2007*). CDK inhibition begins at the metaphase to anaphase transition when APC/C-Cdc20 targets all S phase cyclins and a fraction of mitotic cyclins for destruction. Then, in a second step, APC/C bound to a specificity factor related to Cdc20 - Cdh1 - brings about the complete degradation of mitotic cyclins and hence complete mitotic CDK inactivation. This results in mitotic exit. APC/C-Cdh1 is aided in mitotic CDK inactivation by the CDK inhibitor Sic1 (*Wäsch and Cross, 2002*).

APC/C-Cdh1 and Sic1 activity are controlled by the essential protein phosphatase Cdc14. Cdc14 dephosphorylates Cdh1 and Sic1 thereby activating them (*Taylor et al., 1997*; *Jaspersen et al., 1998*; *Visintin et al., 1998*; *Zachariae et al., 1998*). Given the central role of Cdc14 in control of exit from mitosis, it is not surprising that the phosphatase is tightly regulated. For most of the cell cycle Cdc14 is bound to its inhibitor Cfi1/Net1 in the nucleolus (*Shou et al., 1999*; *Visintin et al., 1999*). During anaphase the phosphatase dissociates from its inhibitor spreading through the nucleus and cytoplasm to dephosphorylate many proteins including Sic1 and Cdh1.

Two signaling pathways have been identified that control the interaction between Cdc14 and its inhibitor. The Cdc14 Early Anaphase Release (FEAR) network releases Cdc14 from its inhibitor in the nucleolus upon anaphase entry (*Stegmeier et al., 2002*; *Pereira and Schiebel, 2003*; *D'Amours et al., 2004*; *Lavoie et al., 2004*; *Sullivan et al., 2004*; *Higuchi and Uhlmann, 2005*; *Khmelinskii et al., 2007*; *Woodbury and Morgan, 2007*; *Roccuzzo et al., 2015*). The FEAR network only transiently releases Cdc14 from its inhibitor. Cdc14 will return to the nucleolus unless the mitotic exit network (MEN) becomes active. Activation of the MEN in late anaphase causes the sustained release of Cdc14 from the nucleolus, resulting in exit from mitosis (*Visintin et al., 1998*; *Visintin et al., 1999*; *Jaspersen et al., 1999*; *Lee et al., 2001*).

The FEAR network is comprised of some of the same proteins that control chromosome segregation at the onset of anaphase (*Figure 1A*). Separase not only cleaves cohesins, the protease also associates with Slk19 which leads to the phosphorylation of the Cdc14 inhibitor Cfi1/Net1 by an unknown mechanism (*Sullivan and Uhlmann, 2003*; *Yellman and Burke, 2006*; *Queralt et al., 2006*; *Wang and Ng, 2006*; *Azzam et al., 2004*). Cfi1/Net1 phosphorylation combined with activation of the FEAR network protein Spo12 then leads to Cdc14 release from its inhibitor in the nucleolus. While Cdc14 released from Cfi1/Net1 by the FEAR network has many roles in facilitating anaphase chromosome segregation (*Rock and Amon, 2009*), the network is not essential for exit from mitosis (*D'Amours et al., 2004*; *Yoshida et al., 2002*; *Pereira et al., 2002*). Inactivation of the FEAR network only causes a 20 min delay in exit from mitosis (*Stegmeier et al., 2002*).

The Mitotic Exit Network (MEN) is activated during anaphase and promotes the sustained release of Cdc14 from the nucleolus. Released Cdc14 then triggers complete CDK inactivation and hence exit from mitosis. The MEN is essential. Cells lacking MEN function arrest in late anaphase failing to exit from mitosis (*Shirayama et al., 1994*; *Bardin et al., 2000*; *Geymonat et al., 2002*). The signaling pathway resembles a Ras-like GTPase signal transduction cascade and is comprised of the GTPase Tem1, the protein kinases Cdc15 and Dbf2-Mob1, and the signaling scaffold Nud1 (see Figure 7A for a MEN diagram; reviewed in *Stegmeier and Amon, 2004*). MEN components are scaffolded onto spindle pole bodies (SPBs), the yeast equivalents of centrosomes, by Nud1 (*Valerio-Santiago and Monje-Casas, 2011*). Tem1 recruits Cdc15 to SPBs during anaphase (*Visintin and Amon, 2001*; *Rock and Amon, 2011*). Cdc15 then activates Dbf2-Mob1 (*Rock et al., 2013*). Active

Dbf2-Mob1 promotes the release of Cdc14 from the nucleolus (*Mohl et al., 2009*; *Manzoni et al., 2010*).

One function of the MEN is to ensure that sustained Cdc14 activation only occurs when the nucleus moves into the bud during anaphase. Asymmetric localization of MEN activators and inhibitors causes the MEN to react to spindle position. Kin4, which activates the Tem1 GTPase activating protein (GAP) complex Bub2-Bfa1 and hence maintains Tem1 in the GDP bound form (*Shirayama et al., 1994*; *Geymonat et al., 2002*; *Li, 1999*; *Bloecher et al., 2000*) localizes to the mother cell (*D'Aquino et al., 2005*; *Pereira and Schiebel, 2005*; *Maekawa et al., 2007*). Lte1, which inhibits Kin4 and hence activates Tem1, localizes to the bud (*Bertazzi et al., 2011*; *Falk et al., 2011*; *Falk et al., 2016a*). Thus, Kin4 and Lte1 create a MEN inhibitory zone in the mother cell and a MEN activating zone in the bud, respectively (*Figure 1B*) (*Bardin et al., 2000*; *Pereira et al., 2000*; *Chan and Amon, 2010*; *Falk et al., 2016b*). This setup ensures that the MEN can only be activated when a SPB carrying the MEN GTPase translocates into the bud during anaphase, hence making the MEN responsive to spindle position.

While an appealing model for how cells sense spindle position, this zone model of MEN activation fails to explain how mitotic exit is coordinated with cell cycle progression. It has previously been observed that when cells experience a prolonged metaphase arrest the nucleus moves into the bud (*Palmer et al., 1992*), yet these cells do not exit from mitosis. Why does the MEN not promote exit from mitosis in these metaphase arrested cells despite a MEN GTPase-bearing SPB moving into the bud (*Figure 1B*)? We show here that high mitotic CDK activity prevents MEN activation in metaphase, even when a MEN GTPase-bearing SPB moves into the bud. We further show that APC/C-Cdc20 dependent cyclin degradation and activation of Cdc14 by the FEAR network lowers mitotic CDK activity to permit MEN activation upon movement of the SPB into the bud. Surprisingly, antagonism by CDKs does not inhibit the MEN GTPase Tem1, but rather causes phosphorylation of downstream components of the MEN, the protein kinases Cdc15 and Mob1-Dbf2. CDK inhibition of the MEN kinase cascade prevents the feedback loops that would otherwise be initiated upon movement of a MEN GTPase-bearing SPB into the bud. Our study shows that the MEN integrates spatial and temporal signals at different steps of the signaling pathway, demonstrating logical 'AND gate' control of the pathway. This signal integration logic ensures that exit from mitosis only occurs after anaphase entry AND after partitioning of the genetic material between mother and daughter cell. Utilizing multi-point regulation explains how GTPase pathways perform signal integration and provides a rationale for why signaling pathways are multi-step processes.

## Results

### MEN activity is confined to anaphase in the strain W303

The MEN promotes exit from mitosis. This activity of the MEN is confined to anaphase. During this cell cycle stage movement of a SPB into the bud leads to the association of the MEN kinase Cdc15 with SPBs where it phosphorylates the scaffold Nud1 (*Rock et al., 2013*). This promotes the association of Dbf2-Mob1 with SPBs, resulting in activation of the kinase (*Visintin and Amon, 2001*). Dbf2-Mob1 then activates Cdc14 to trigger exit from mitosis.

A recent study reported that the MEN kinases Cdc15 and Dbf2-Mob1 are active prior to anaphase and prior to a SPB moving into the bud, but that this activity cannot be measured by conventional kinase activity assays (*Hotz and Barral, 2014*). This metaphase-specific MEN activity was reported to affect spindle positioning (*Hotz et al., 2012a*; *Hotz et al., 2012b*) by controlling the localization of the spindle positioning factor Kar9. When the MEN was inactivated, Kar9 localization to astral microtubules emanating from the old, daughter bound SPB was impaired, resulting in the aberrant inheritance of the newly synthesized SPB by the daughter cell. These results raised the interesting possibility that low levels of SPB-localized MEN activity promote spindle positioning during metaphase but are insufficient to bring about exit from mitosis. A key question arising from this observation is how MEN activity is controlled in a manner that allows the pathway to fulfill its spindle orientation function in metaphase, yet does not promote exit from mitosis even when a MEN-bearing SPB transiently moves into the bud during metaphase.

To address this question, we examined Kar9 localization in cells either harboring a temperature sensitive (*cdc15-2*) or an analog sensitive (*cdc15-as1*) allele of the MEN kinase *CDC15*. Kar9

association with astral microtubules is asymmetric. Kar9 preferentially binds to microtubules that emanate from the old SPB, that is destined to move into the bud during anaphase (*Liakopoulos et al., 2003*). *Hotz et al. (2012a)* reported that Kar9 localization becomes symmetric when the MEN is inhibited, associating almost equally well with microtubules emanating from both, the new and old SPB. We found that Kar9 localization was not affected by MEN inhibition. Kar9 localization in *cdc15-2* and *cdc15-as1* cells was indistinguishable from that of wild-type cells (*Figure 1—figure supplement 1A–M*). Because Kar9 localization is highly dynamic we also performed short-term (100 s, 10 s per frame; *Figure 1—figure supplement 1I–J*) and long-term imaging studies (4 hr, 5 m per frame; *Figure 1—figure supplement 1K–M*). Inactivation of the MEN did not affect Kar9 localization in either experimental set up.

The changes in Kar9 localization found in MEN mutants was reported to cause spindle pole body inheritance defects (*Hotz et al., 2012a*). In wild-type cells, the old SPB is inherited by the daughter cell, whereas the newly synthesized SPB remains in the mother cell (*Pereira et al., 2001*). In MEN mutants this stereotypic inheritance pattern was reported to be perturbed (*Hotz et al., 2012a*). To measure SPB inheritance we fused the SPB protein Spc42 to the slow *maturing* mCherry fluorophore. This fusion makes the old SPB appear brighter than the newly synthesized one, allowing us to follow SPB inheritance (*Pereira et al., 2001*) (*Figure 1—figure supplement 1O*). As previously reported, we found that deletion of *KAR9* randomized SPB inheritance (*Pereira et al., 2001*) (*Figure 1—figure supplement 1N*). However, inactivation of *CDC15* using the *cdc15-as1* allele, did not affect SPB inheritance (*Figure 1—figure supplement 1N*). We conclude that in our strain background, W303, there is no evidence for MEN activity prior to anaphase onset. We propose that strain background differences could be the reason why our results differ from those reported by *Hotz et al. (2012a)*. Based on a lack of evidence for MEN activity outside of anaphase in our strain background, we will define 'MEN activity' by its ability to promote mitotic exit throughout the remainder of the paper.

## APC/C-Cdc20 activity is required for MEN activation

The nucleus has been observed to make transient excursions into the bud prior to anaphase (*Palmer et al., 1992*; *Yeh et al., 1995*). This is particularly evident in APC/C mutants that arrest in metaphase. Cells harboring a temperature sensitive allele of the APC/C subunit encoding gene *CDC23* (*cdc23-1*) arrest in metaphase. During the arrest, 97% of cells move part or the entirety of the undivided nucleus into the bud. To determine whether MEN activation occurs in *cdc23-1* cells in which the nucleus translocated into the bud, we analyzed Mob1 localization. In wild-type cells, Mob1 associates with SPBs when the MEN is activated during anaphase (*Figure 1C*; *Figure 1—figure supplement 2*) (*Rock et al., 2013*). Following MEN activation, Cdc14 is released into the cytoplasm (*Figure 1D Figure 1—figure supplement 2*). Cdc14 releae causes exit from mitosis and entry into the next cell cycle ensues, as judged by cells forming a new bud (referred to as rebudding; *Figure 1E*).

In metaphase-arrested *cdc23-1* mutants neither MEN activation nor exit from mitosis occurred. SPB movement into the bud caused only weak, inconsistent Mob1 association with SPBs, Cdc14 was not released from the nucleolus, and cells failed to enter the next cell cycle (*Figure 1C–E*; *Figure 1—figure supplement 2*). It is, however, noteworthy that some Mob1 associated weakly with SPBs when the nucleus moved into the bud in *cdc23-1* mutants (*Figure 1C*; *Figure 1—figure supplement 2*). This observation suggests that the MEN retained at least some ability to respond to spindle position. Yet, the low level of Mob1 association with SPBs combined with lack of Cdc14 release from the nucleolus and lack of rebudding indicates that MEN activation is suppressed during metaphase. We conclude that movement of a SPB into the bud causes MEN activation only when it occurs during anaphase.

## Spindle elongation is not sufficient to activate the MEN

Which aspect of anaphase is critical for MEN activation? Previous work has established that APC/C-Cdc20 has two essential functions in promoting anaphase onset: cyclin degradation and Securin degradation (*Figure 1A*) (*Shirayama et al., 1999*). Cyclin degradation reduces CDK activity during anaphase. Securin degradation activates the FEAR network and causes cohesin cleavage, which in turn leads to spindle elongation and chromosome segregation. Any or all of these anaphase events could be required for MEN activation during anaphase.

We first tested the role of anaphase spindle elongation in MEN activation. To render spindle elongation independent of APC/C-Cdc20 activity we introduced a temperature sensitive cohesin allele known as *mcd1-1* into *cdc23-1* cells. This double mutant undergoes spindle elongation during a metaphase arrest because sister chromatids are no longer held together due to the loss of cohesin function. Inactivation of cohesins activates the spindle assembly checkpoint (SAC) (*Severin et al., 2001*). To prevent interference of the SAC with anaphase events we deleted the SAC gene *MAD1*. Analysis of the *cdc23-1, mcd1-1, mad1Δ* strain revealed that spindle elongation was not sufficient to activate the MEN in metaphase. Mob1 only weakly associated with SPBs, Cdc14 was not released from the nucleolus, and cells failed to enter the next cell cycle (*Figure 1C–E*; *Figure 1—figure supplement 2*).

Similar results were obtained when we inactivated the APC/C-Cdc20 by depleting Cdc20. We arrested cells in G1 using alpha factor pheromone and synchronously released them into the cell cycle under conditions that inactivated cohesin and depleted Cdc20. We measured MEN activity by assaying for Cdc15 activity. The protein kinase phosphorylates Nud1 on Threonine 78, which can be detected using a phospho-specific antibody (*Figure 1F*) (*Rock et al., 2013*). In *mcd1-1 mad1Δ* control cells Nud1 T78 phosphorylation occurred during anaphase. Mitotic exit followed as judged by the disassembly of the anaphase spindle. In cells depleted for Cdc20, T78 phosphorylation was greatly delayed and diminished, irrespective of whether or not spindle elongation occurred (*Figure 1F*). Furthermore, cells failed to exit from mitosis as judged by the persistence of anaphase spindles in cells. We conclude that movement of a SPB into the bud is a prerequisite for MEN activation but spindle elongation is not.

## Cdc14 activated by the FEAR network contributes to MEN activation in anaphase

Degradation of Securin by APC/C-Cdc20 not only triggers cohesin cleavage and hence spindle elongation, it also leads to activation of the FEAR network (*Figure 1A*). The FEAR network transiently releases Cdc14 from the nucleolus during early anaphase. Previous studies showed that Cdc14 release from the nucleolus mediated by the FEAR network is required for the timely activation of the MEN (*Stegmeier et al., 2002*; *Yoshida et al., 2002*; *Pereira et al., 2002*). In the absence of FEAR network function MEN activation is delayed by approximately 15–20 min. We confirmed these previous findings using the *mad1Δ mcd1-1* experimental set-up, because the use of a cohesin mutant ensured delivery of a MEN bearing SPB into the bud to allow for activation of Tem1 by Lte1.

Inactivation of the FEAR network led to a highly consistent 15 min delay in MEN activation (*Figure 2*). Upon release from a pheromone induced G1 arrest, overexpression of a non-degradable form of Pds1 (*GAL-PDS1dBΔ*) caused a 15 min delay in Nud1T78 phosphorylation and anaphase spindle break-down (*Figure 2A*). In contrast, arrest of cells in metaphase using the microtubule depolymerizing drug nocodazole completely prevented MEN activation and mitotic exit. The same result was obtained with an additional assay for MEN activity: an IP-kinase assay to measure activity of the Dbf2-Mob1 kinase (*Figure 2B*) (*Visintin and Amon, 2001*). A similar 15 min delay in Nud1 T78 phosphorylation, Dbf2 kinase activation, and anaphase spindle breakdown was observed in cells harboring a temperature sensitive allele of Separase (*esp1-1*) and in two other FEAR network mutants: *slk19Δ* and *spo12Δ bns1Δ* (*Figure 2C,D*; *Figure 3A-D*; *Figure 3—figure supplement 1*).

To determine whether the FEAR network promotes MEN activity by activating Cdc14, we measured MEN activity in cells carrying the temperature sensitive *cdc14-3* allele. *cdc14-3* cells synchronously released from a pheromone-induced G1 arrest exhibited a 30 min delay in Nud1 T78 phosphorylation and Dbf2 kinase activation (*Figure 3E,F*). We conclude that activation of the FEAR network at the metaphase – anaphase transition contributes to the timely activation of the MEN.

## High CDK activity inhibits MEN activation

Lack of FEAR network activity in metaphase cannot explain why MEN activation does not occur when a MEN bearing SPB moves into the bud. This is evident when comparing MEN activity of Cdc20-depleted cells with that of cells expressing a non-degradable version of Securin (*Figure 4A*). Cdc20-depleted cells neither activated the MEN as judged by a lack of Nud1T78 phosphorylation nor exited mitosis as judged by the persistence of anaphase spindles. In contrast, cells in which FEAR network activity was inhibited by expressing a non-degradable version of Pds1 delayed MEN

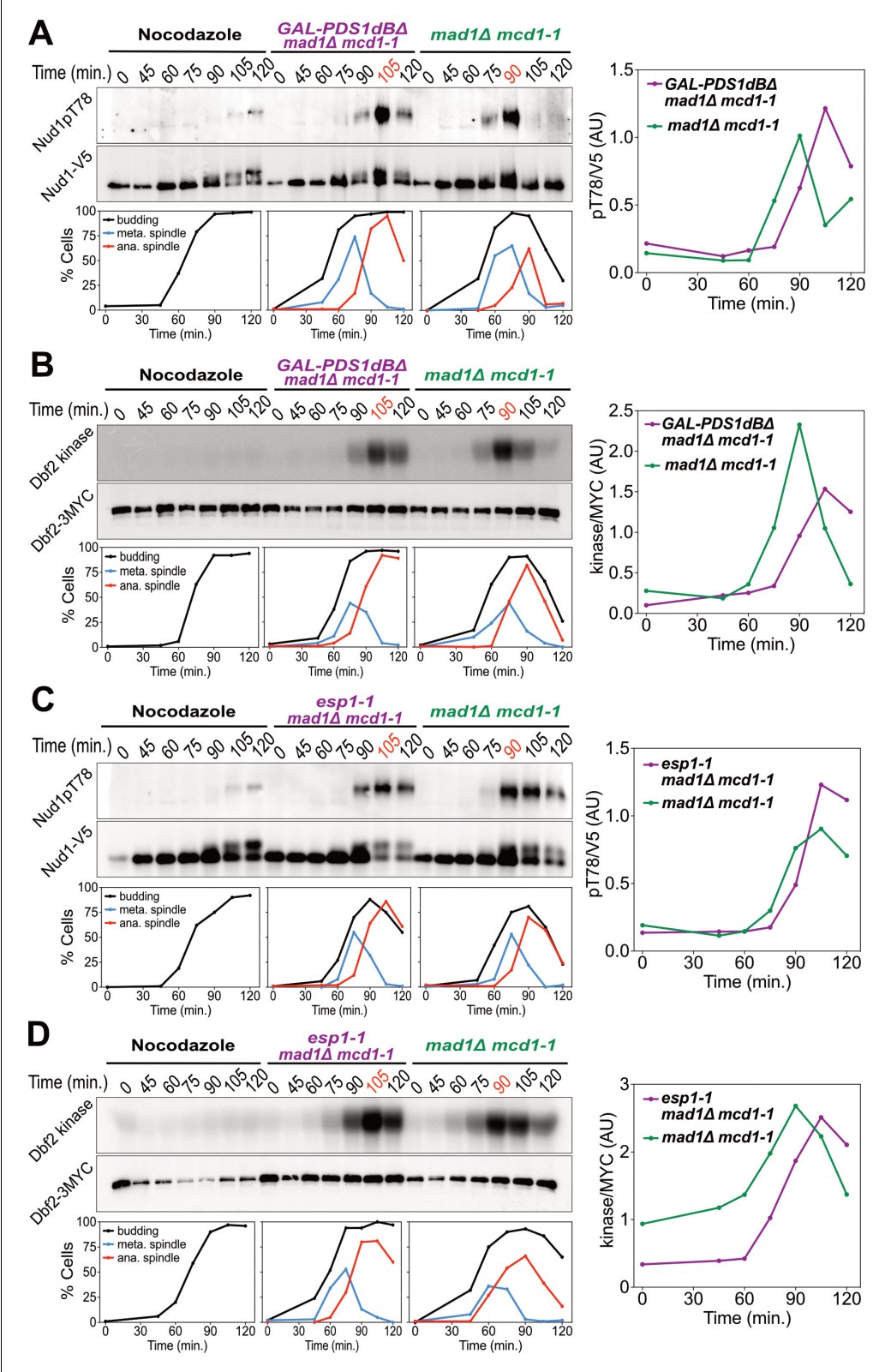

**Figure 2.** Securin degradation and Separase function promote MEN activation. (**A, B**) *GAL-PDS1dBΔ mad1Δ mcd1-1* and *mad1Δ mcd1-1* cells containing (**A**) *NUD1-3V5* (A37904 and A37739) or (**B**) *DBF2-3MYC* (A37497 and A37523) were arrested in G1 with α-factor pheromone at room temperature in YEP medium containing raffinose. After 3 hours cells were released into pheromone-free YEP medium containing raffinose and galactose at 34°C. A control sample was released from the G1 arrest into medium containing nocodazole (15 μg/ml) to arrest cells in metaphase. (**C, D**) *Figure 2 continued on next page*

*Figure 2 continued*

*esp1-1 mad1Δ mcd1-1* and *mad1Δ mcd1-1* cells containing (**C**) *NUD1-3V5* (A38079 and A37739) or (**D**) *DBF2-3MYC* (A37500 and A37523) were arrested in G1 with α-factor pheromone at room temperature in YEPD medium. After 3 hours cells were released into pheromone-free YEPD medium at 37°C. A control sample was released from the G1 arrest into medium containing nocodazole. (**A–D**) Percentage of cells with buds (black), metaphase spindles (blue), and anaphase spindles (red) was determined at the indicated times. (**A, C**) Nud1 T78 phosphorylation and total Nud1 levels were determined. (**B, D**) Dbf2 kinase activity and Dbf2 protein levels were determined.

DOI: https://doi.org/10.7554/eLife.41139.005

activation and exit from mitosis by only 20 min. This result demonstrates that the APC/C-Cdc20 must have an additional role in MEN activation separate from activation of the FEAR network.

We hypothesized that APC-Cdc20's function in promoting the degradation of S phase and mitotic cyclins could explain why MEN activity is restricted to anaphase. To test this hypothesis we analyzed the consequences of overexpressing a non-degradable form of Clb2 (Clb2dBΔ). We found that over-expression of Clb2dBΔ prevented MEN activation during anaphase as judged by reduced Mob1 association with SPBs (*Figure 4B*; *Figure 4—figure supplement 1*). Exit from mitosis also did not occur as judged by the inability of cells to form a new bud (*Figure 4C*). Combining overexpression of Clb2dBΔ with the *cdc14-3* allele completely abolished Mob1 binding to SPBs (*Figure 4B*; *Figure 4—figure supplement 1*). These data demonstrate that high CDK activity is sufficient to inhibit the MEN.

## APC/C-Cdc20 degradation of Clb2 is essential for MEN activation

The observation that high mitotic CDK activity in anaphase is sufficient to prevent MEN activation suggested that high mitotic CDK activity prevents MEN activation during metaphase. To test this possibility, we examined the consequences of lowering mitotic CDK activity during metaphase. We arrested cells in metaphase by inhibiting APC/C activity using the temperature sensitive *cdc20-1* or *cdc23-1* mutations. To lower mitotic CDK activity we introduced the temperature sensitive *clb2-VI* allele into these strains and examined exit from mitosis. To measure exit from mitosis we analyzed Cdc3 localization. Cdc3 is a component of the septin ring that dissociates from the structure in response to mitotic CDK inactivation during exit from mitosis (*Kim et al., 1991*). Loss of Cdc3 from the bud neck is thus a sign that mitotic CDKs have been completely inactivated and exit from mitosis has occurred.

Analysis of Cdc3 localization led to the discovery that exit from mitosis depends on both the state of mitotic CDK activity and spindle position. *cdc20-1* single mutants arrested in metaphase and did not exit from mitosis as judged by persistent Cdc3 localization at the bud neck (*Figure 5A,D*). Next, as a positive control, we examined the effects of deleting *BUB2*, which encodes a subunit of Tem1's two-component GAP. Deletion of *BUB2* leads to premature MEN activation and, after a significant delay, exit from mitosis in metaphase-arrested cells (reviewed in *Gardner and Burke, 2000*). We confirmed this result and further found that mitotic exit occurred in a partially spindle position independent manner (*Figure 5A*). 78% of cells in which the spindle migrated into the bud exited from mitosis. In mother cells this number was smaller. Only 36% of cells in which the spindle remained in the mother cell exited mitosis. Interestingly, mitotic exit was delayed in both categories of cells. Time from the onset of budding to exit from mitosis took on average 283 min in *cdc20-1 bub2Δ* cells compared to 86 min in wild-type cells (*Figure 5B*). This delay confirms that a MEN inhibitor, presumably CDK activity, prevents MEN activation in a manner that is GAP independent.

We also noted that some *cdc20-1 bub2Δ* cells elongated their spindles prior to cytokinesis (*Figure 5C*). This observation suggests that activation of APC/C-Cdh1 by the MEN not only causes mitotic cyclin degradation but that APC/C-Cdh1 can also induce Securin degradation. Consistent with this interpretation is the observation that deletion of *BUB2* neither caused exit from mitosis nor spindle elongation in *cdc23-1* mutants (*Figure 5A–D*). In this mutant both forms of the APC are inactive and neither mitotic cyclins nor Securin can be degraded.

Next, we examined the effects of lowering mitotic CDK activity in the two APC/C mutants. Inactivating the mitotic cyclin Clb2 using the *clb2-VI* allele caused mitotic exit in *cdc20-1* mutants (*Figure 5A,D*). Importantly, this exit from mitosis depended on spindle position. Mitotic exit only occurred in cells in which the nucleus had moved into the bud during the arrest. A similar dependence was observed when analyzing the ability of *cdc23-1* cells to exit from mitosis (*Figure 5A*).

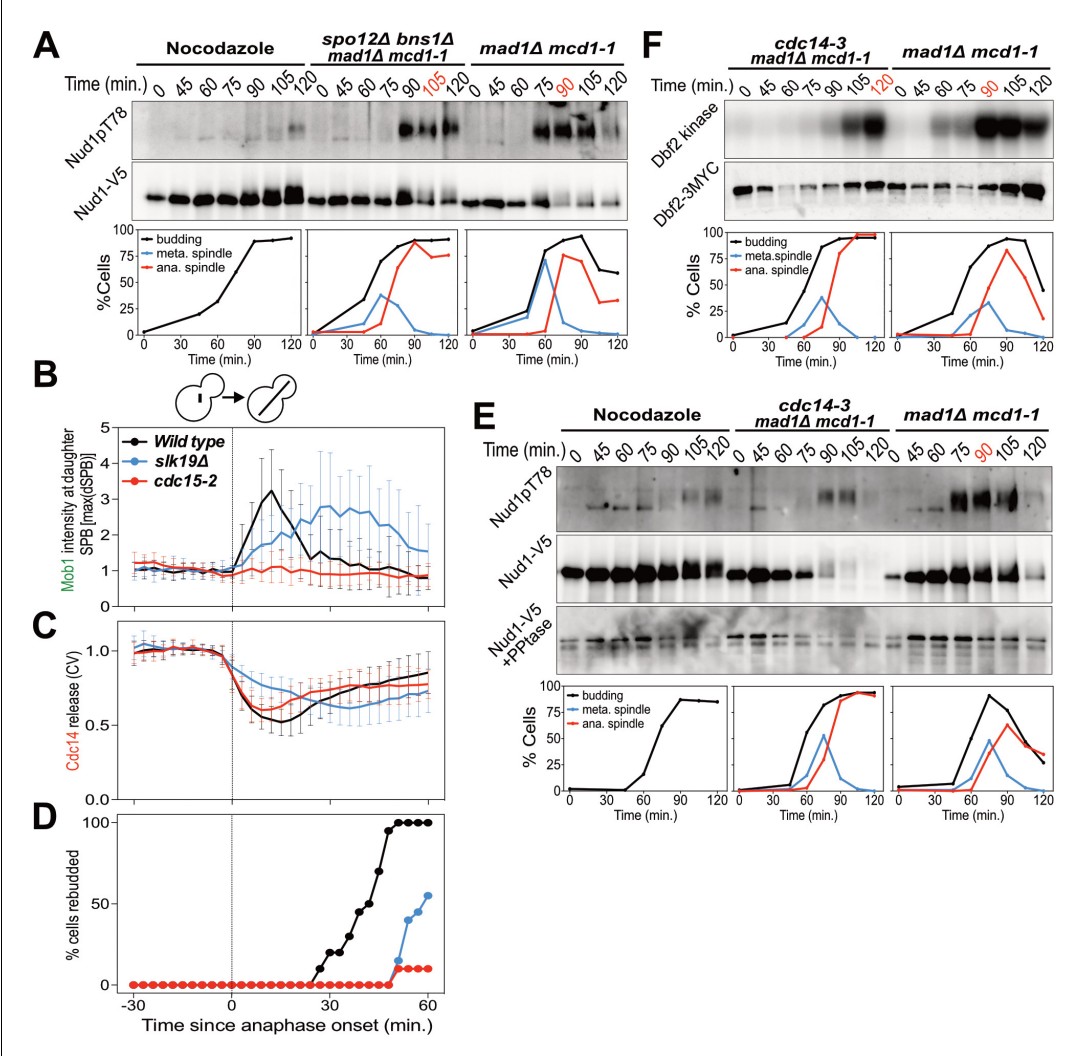

**Figure 3.** The FEAR network promotes MEN signaling by activating Cdc14. (**A**) *spo12Δ bns1Δ mad1Δ mcd1-1* (A38127) and *mad1Δ mcd1-1* (A37739) cells containing *NUD1-3V5* were arrested in G1 with α-factor pheromone at room temperature. (**E, F**) *cdc14-3 mad1Δ mcd1-1* and *mad1Δ mcd1-1* cells containing (**E**) *NUD1-3V5* (A38112 and A37739) or (**F**) *DBF2-3MYC* (A37714 and A37523) were arrested in G1 with α-factor pheromone at room temperature. (**A, E, F**) After 3 hours cells were released into pheromone-free medium at 37°C. A control sample was released from the G1 arrest into medium containing nocodazole. The percentage of cells with buds (black), metaphase spindles (blue), and anaphase spindles (red) was determined at the indicated times. (**A, E**) Nud1 T78 phosphorylation and total Nud1 levels were determined. Note, Nud1 is hyperphosphorylated in *cdc14-3* cells. Samples were therefore treated with Lambda Phosphatase to detect the protein (Nud1-V5 + PPtase, Western). (**F**) Dbf2 associated kinase activity and Dbf2 protein levels were determined. (**B, C, D**) *Wild type* (A39323), *slk19Δ* (A39378), and *cdc15-2* (A39353) cells containing *MOB1-eGFP*, *SPC42-mCherry*, and *CDC14-tdTomato* were grown at 34°C and imaged every 3 min for 4 hr. Curves show mean and stdev. (n = 20 cells; see **Figure 3—figure supplement 1** for individual traces). Data were normalized to the average intensity measured during the 15 min prior to anaphase. Data were centered at the first frame where anaphase onset (spindle >3 μm) was detected. This centered timepoint was designated *t* = 3 min. (**B**) Mob1-eGFP intensity at the daughter bound SPB[max(dSPB)]. (**C**) Cdc14 coefficient of variation (CV = stdev/mean) within the cell. (**D**) Percent cells that rebudded, indicating exit from mitosis and entry into the next cell cycle occurred.

DOI: https://doi.org/10.7554/eLife.41139.006

The following figure supplement is available for figure 3:

**Figure supplement 1.** Deletion of *SLK19* delays MEN activation and Cdc14 release from the nucleolus.

DOI: https://doi.org/10.7554/eLife.41139.007

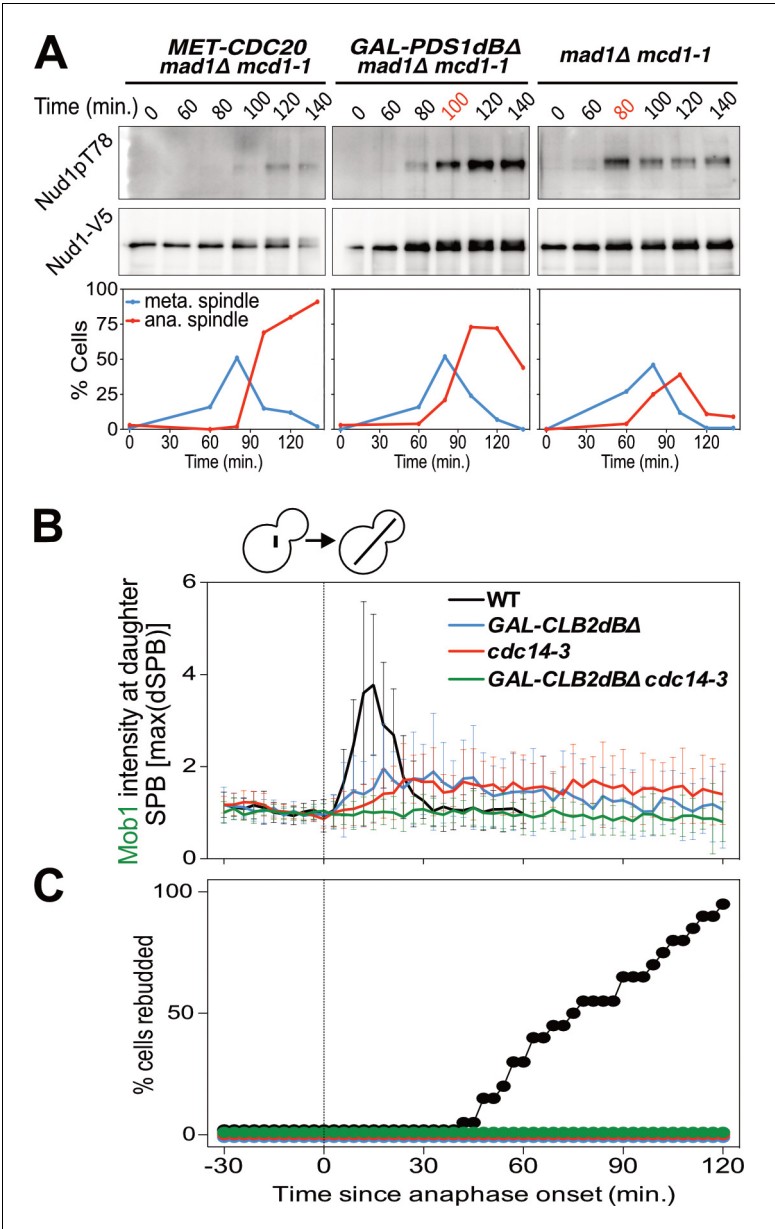

**Figure 4.** Mitotic CDKs antagonize the MEN. (**A**) *MET-CDC20 mad1Δ mcd1-1* (A37907), *GAL-PDS1dBΔ mad1Δ mcd1-1* (A37904), and *mad1Δ mcd1-1* (A37739) cells containing *NUD1-3V5* were arrested in G1 with α-factor pheromone at room temperature in synthetic medium lacking methionine and supplemented with raffinose. After 3 hours cells were released into pheromone-free YEP medium containing raffinose, galactose and 8 mM methionine at 34°C. Methionine was re-added every hour. The percentage of cells with buds (black), metaphase spindles (blue), and anaphase spindles (red) as well as Nud1 T78 phosphorylation and total Nud1 levels were determined at the indicated times. (**B,C**) *Wild type* (A33715), *GAL-CLB2dBΔ* (A39542), *cdc14-3* (A39541), and *GAL-CLB2dBΔ cdc14-3* (A39540) cells containing *MOB1-eGFP* and *SPC42-mCherry* were grown at 34°C in SC medium containing raffinose and galactose and imaged every 3 min for 5 hr. Curves show mean and stdev. (n = 20 cells; see *Figure 4—figure supplement 1* for individual traces). Data were normalized to the average intensity measured during the 15 min prior to anaphase. Data were centered at the first frame where anaphase onset (spindle >3 μm) was detected. This centered timepoint was designated *t* = 3 min. (**B**) Mob1-eGFP intensity at daughter bound SPB [max(dSPB)] was determined. (**C**) Percent cells that rebudded, indicating that exit from mitosis and entry into the next cell cycle had occurred.

DOI: https://doi.org/10.7554/eLife.41139.008

The following figure supplement is available for figure 4:

*Figure 4 continued on next page*

*Figure 4 continued*

**Figure supplement 1.** Clb2 degradation and Cdc14 released from the nucleolus by the FEAR network activate the MEN.

DOI: https://doi.org/10.7554/eLife.41139.009

*cdc20-1* mutants were more likely to undergo mitotic exit than *cdc23-1* mutants (*Figure 5A*), most likely because *cdc20-1* mutants can still activate the APC/C-Cdh1, whereas *cdc23-1* mutants cannot. We note that unlike in *cdc20-1 bub2Δ* mutants spindle elongation did not occur in either *cdc20-1 clb2-VI* or *cdc23-1 clb2-VI* mutants. Spindle elongation requires mitotic CDK activity (*Rahal and Amon, 2008*). The *clb2-VI* allele likely lowers mitotic CDK activity below the level necessary to accomplish this task.

To determine whether the exit from mitosis we observed in *cdc20-1 clb2-VI* mutants was due to activation of the MEN, we examined MEN activity in this strain. As mitotic exit did not occur in all cells we developed a live cell assay for MEN activation that was compatible with co-observation of Cdc3 localization. To selectively measure MEN activity we took advantage of the finding that the MEN kinase Dbf2-Mob1 phosphorylates the NLS of Cdc14, increasing cytoplasmic Cdc14 when the MEN is active (*Mohl et al., 2009*). We fused the C-terminus of Cdc14, which contains the NLS, to a fluorescent reporter ($NLS_{Cdc14}$). Release of $NLS_{Cdc14}$ from the nucleus into the cytoplasm depended on MEN activity, validating this novel MEN assay (*Figure 5—figure supplement 1A–C*). Furthermore, in the FEAR network mutant *slk19Δ*, export of $NLS_{Cdc14}$ from the nucleus was delayed by 20 min, which is consistent with the 20 min delay in MEN activation observed in *slk19Δ* mutants. (*Figure 5—figure supplement 1A–E*). Most importantly, the analysis of *slk19Δ* mutants demonstrated that the $NLS_{Cdc14}$ localization assay specifically reports on MEN-dependent control of Cdc14. When we compared the release of full-length Cdc14 from the nucleolus with export of the $NLS_{Cdc14}$ from the nucleus we found that the time difference between the two was smaller in FEAR network mutants than in wild-type cells (*Figure 5—figure supplement 1F*). This is due to the fact that release of full length Cdc14 from the nucleolus only depends on MEN activity in FEAR network mutants, delaying its release. As release of the $NLS_{Cdc14}$ is not mediated by the FEAR network, the time delay between the two events is smaller. We conclude that $NLS_{Cdc14}$ localization specifically reports on MEN activity.

*cdc20-1* single mutants arrested in metaphase did not release $NLS_{Cdc14}$ into the cytoplasm when a SPB translocated into the bud which is consistent with their failure to disassociate Cdc3 from the bud neck and to exit from mitosis (*Figure 5E,F*). Deletion of *BUB2* caused release of $NLS_{Cdc14}$ into the cytoplasm in metaphase-arrested cells (*Figure 5E,F*). In *cdc20-1 clb2-VI* mutants in which exit from mitosis occurred (Cdc3 delocalized from the bud neck), we also observed translocation of $NLS_{Cdc14}$ from the nucleus into the cytoplasm (*Figure 5E,F*), indicating that exit from mitosis in this strain was preceded by MEN activation. Indeed, we found that spindle position dependent exit from mitosis in *cdc20-1 clb2-VI* cells required MEN activity. *cdc20-1 clb2-VI cdc15-2* triple mutants failed to exit mitosis upon translocation of a SPB into the bud (*Figure 5A*). Together, our results demonstrate that MEN activation requires two signals: (1) movement of a MEN GTPase-bearing SPB into the bud and (2) a lowering of mitotic CDK activity by APC/C-Cdc20 at the onset of anaphase.

## DNA damage inhibits the MEN by additional mechanisms

In budding yeast, activation of the DNA damage checkpoint causes cell cycle arrest in metaphase (*Agarwal et al., 2003*). We therefore asked whether high mitotic CDK activity also prevented MEN activation in this arrest. We used three methods to trigger activation of the DNA damage checkpoint: the temperature sensitive *cdc13-1* and *cdc9-1* alleles as well as hydroxyurea (HU). *CDC13* is required for telomere function. In its absence large amounts of single stranded DNA accumulate at telomeres, which causes activation of the DNA damage checkpoint (*Weinert et al., 1994*; *Garvik et al., 1995*; *Lin and Zakian, 1996*). *CDC9* mutants fail to join Okazaki fragments, causing double strand breaks to form which in turn activate the checkpoint (*Johnston and Nasmyth, 1978*; *Schiestl et al., 1989*). HU inhibits ribonucleotide reductase thus causing nucleotide depletion and replication stress.

As in our analysis of *cdc20-1* and *cdc23-1* mutants we first examined the effects of deleting *BUB2* on the DNA damage-induced metaphase arrest. Deletion of *BUB2* increased the percentage of cells

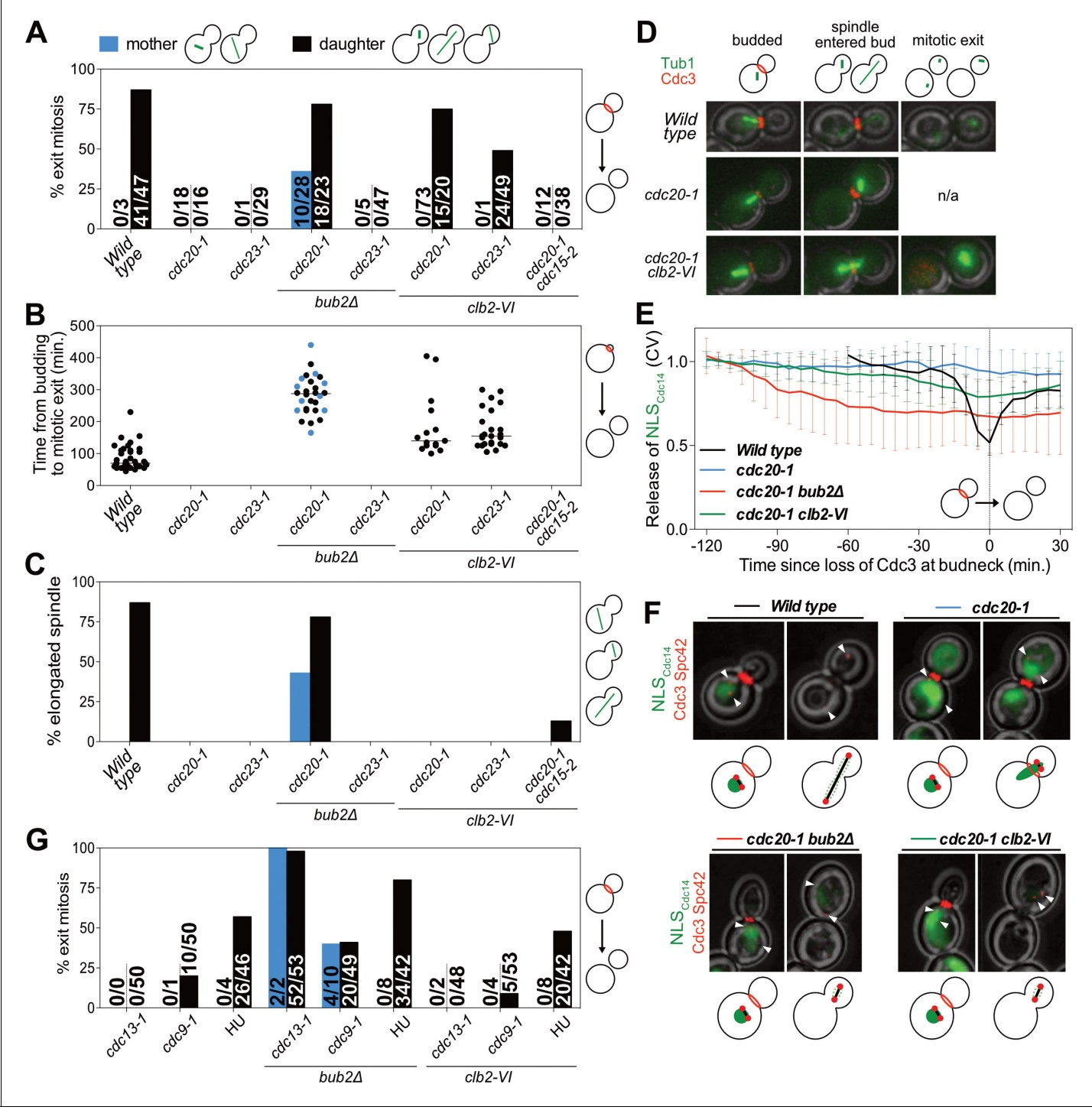

**Figure 5.** Mitotic CDKs prevent MEN activation in metaphase. (**A–D, G**) The indicated strains containing *mCherry-Cdc3* and *GFP-Tub1* were grown at 37°C and imaged every 5 min for 5 (*WT*) or 8 (remaining strains) hours. (**A**) Spindle position was monitored and cells were apportioned by whether the spindle had entered the daughter cell (daughter) or remained in the mother cell (mother). Mitotic exit was measured by loss of Cdc3 from the bud neck. (**B**) Time from bud emergence until mitotic exit for the cells shown in (**A**). (**C**) Percent of cells from (**A**) that elongated their spindles (>3 µm). (**D**) Sample images for (**A–C**). Cdc3 is shown in red, microtubules in green. (**E, F**) *Wild type* (A40293), *cdc20-1* (A40295), *cdc20-1 bub2Δ* (A40296), and *cdc20-1 clb2-VI* (A40292) cells containing *mCherry-Cdc3*, *Spc42-mCherry*, and *NLS$_{Cdc14}$-GFP* were grown at 37°C and imaged every 5 min for 4 (*WT*), or 8 (remaining strains) hours. Curves in (**E**) show mean and stdev. (n = 20 cells; see *Figure 5—figure supplement 2* for individual traces). Data were normalized to the average intensity measured during the initial 20 min of the analysis. Data were centered so that loss of Cdc3 from the bud neck represents t = 0. *cdc20-1* cells do not lose Cdc3 from the bud neck. We therefore analyzed cells for 2.5 hr after the spindle first moved into the bud.
*Figure 5 continued on next page*

*Figure 5 continued*

NLS$_{Cdc14}$-GFP localization was measured as coefficient of variation (CV = stdev/mean) within the cell. Sample images are shown in (**F**). Cdc3 is shown in red, microtubules in green. (**G**) Spindle position was monitored and mitotic exit was measured by loss of Cdc3 from the bud neck. 5 mg/ml hydroxyurea (HU) was used in indicated samples.

DOI: https://doi.org/10.7554/eLife.41139.010

The following figure supplements are available for figure 5:

**Figure supplement 1.** NLS$_{Cdc14}$ is a live-cell MEN activity reporter.

DOI: https://doi.org/10.7554/eLife.41139.011

**Figure supplement 2.** Clb2 degradation and Mob1 dephosphorylation promote MEN activation.

DOI: https://doi.org/10.7554/eLife.41139.012

exiting mitosis in all three DNA damage-inducing conditions (*Figure 5G*). Presumably, as in *cdc20-1 bub2Δ* mutants, deletion of *BUB2* activates the MEN in the DNA damage checkpoint arrest causing APC/C-Cdh1 to promote exit from mitosis.

Interestingly, unlike *cdc20-1 clb2-VI* cells, combining the *clb2-VI* allele with the DNA damage-inducing conditions did not cause more cells to exit from mitosis upon movement of the nucleus into the bud (*Figure 5G*). This finding indicates that mechanisms in addition to high mitotic CDK activity prevent MEN activation in cells arrested in metaphase due to DNA damage. Indeed, previous reports have shown that DNA damage inhibits the MEN by activating the MEN inhibitory GAP complex, Bub2-Bfa1 (*Valerio-Santiago et al., 2013*). Consistently deleting *BUB2* allows *cdc13-1*, *cdc9-1* and HU-treated cells to exit from mitosis (*Figure 5G*). We conclude that at least one mechanism in addition to spindle position and high mitotic CDK activity prevents MEN activation in response to DNA damage. This additional inhibitory mechanism is likely to be activation of the MEN GAP Bub2-Bfa1 by DNA damage.

## Mitotic CDKs inhibit the MEN by phosphorylating components of the kinase cascade

Previous work had established that the MEN kinase Cdc15 has 7 CDK phosphorylation sites (*Jaspersen and Morgan, 2000*); Mob1 has 2 (*Valerio-Santiago et al., 2013*). Mutation of these sites to alanine (*cdc15-7A* and *mob1-2A*) prevents their phosphorylation by mitotic CDKs.

To determine whether CDK phosphorylation of Cdc15 and/or Mob1 was responsible for restricting MEN activity to anaphase we examined the ability of *cdc15-7A* and *mob1-2A* mutants to exit from mitosis in a *cdc20-1* metaphase arrest. We found that 21% of *cdc15-7A cdc20-1* double mutants were able to exit from mitosis (*Figure 6A*). Importantly, as seen when lowering mitotic CDK activity, exit from mitosis depended on movement of a SPB into the bud.

Replacing *MOB1* with *mob1-2A* had a more pronounced effect on exit from mitosis. 52% of *cdc20-1 mob1-2A* cells exited mitosis (*Figure 6A*). Exit from mitosis largely depended on movement of the nucleus into the bud. 52% of cells with the spindle in the daughter cell exited mitosis, whereas only 6% of cells exited mitosis in which the spindle remained in the mother cell (*Figure 6A*). We hypothesize that a transient, unobserved excursion of a MEN-bearing SPB into the bud activated the MEN in this small fraction of cells.

As observed in *cdc20-1 bub2Δ* cells, mitotic exit in *cdc20-1 cdc15-7A* and *cdc20-1 mob1-2A* was slow and proceeded by spindle elongation (*Figure 6B–C*). APC/C-Cdh1 likely triggers Pds1 degradation in these cells. Importantly, exit from mitosis in *cdc20-1 mob1-2A* cells was accompanied by MEN activation. *cdc20-1 mob1-2A* cells that exited mitosis exported NLS$_{Cdc14}$ from the nucleus into the cytoplasm (*Figure 6D*; *Figure 5—figure supplement 2*).

Combining the *cdc15-7A* and *mob1-2A* allele was additive. 80% of *cdc15-7A mob1-2A cdc20-1* triple mutants exited from mitosis upon movement of the nucleus into the bud during the metaphase arrest and cells exited mitosis more rapidly than either of the double mutants (*Figure 6A,B*). In contrast, only 15% of cells where the nucleus remained in the mother cell exited mitosis, presumably due to unobserved, transient excursions by a MEN-bearing SPB into the bud. We conclude that CDK activity inhibits the MEN during metaphase by phosphorylating multiple MEN kinases. At the onset of anaphase, APC/C-Cdc20 attenuates mitotic CDK activity by (1) targeting a fraction of mitotic cyclins for destruction and (2) inducing FEAR-network mediated release of Cdc14 from its

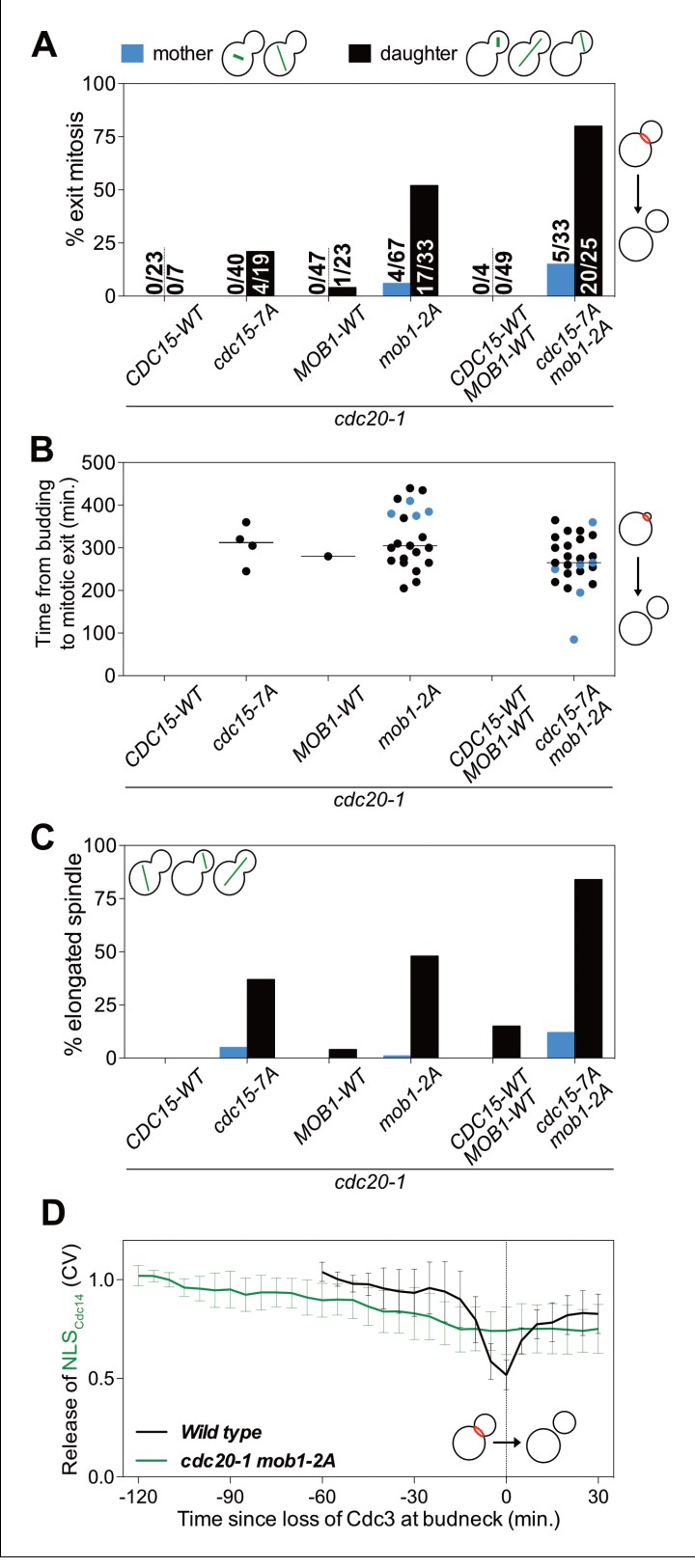

**Figure 6.** CDK activity inhibits the MEN by phosphorylating Cdc15 and Mob1. (**A–C**) The indicated strains containing *mCherry-Cdc3* and *GFP-Tub1* were grown at 37°C and imaged every 5 min for 8 hr. (**A**) Spindle position was monitored and cells apportioned by whether the spindle entered the daughter cell (daughter) or remained in the mother cell (mother). Mitotic exit was measured by loss of Cdc3 from the bud neck. (**B**) Time from bud

*Figure 6 continued on next page*

*Figure 6 continued*

emergence until mitotic exit for the cells shown in (**A**). (**C**) Percent of cells from (**A**) that elongated their spindles (>3 μm). (**D**) *Wild type* (A40293), or *cdc20 mob1-2A* (A40292) cells containing *mCherry-Cdc3*, *Spc42-mCherry*, and *NLS$_{Cdc14}$-GFP* were grown at 37°C and imaged every 5 min for 4 (*WT*), or 8 (*cdc20-1 mob1-2A*) hours. Curves show mean and stdev. (n = 20 cells; see *Figure 5—figure supplement 2* for individual traces). Data were normalized to the average intensity measured during the initial 20 min of the analysis. Data were centered so that loss of Cdc3 from the bud neck represents t = 0. Release of NLS$_{Cdc14}$-GFP was measured as coefficient of variation (CV = stdev/mean) of NLS$_{Cdc14}$-GFP intensity within the cell.

DOI: https://doi.org/10.7554/eLife.41139.013

inhibitor in the nucleolus. These two mechanisms restrict MEN activation to anaphase, preventing premature exit from mitosis.

## Discussion

Our results demonstrate that signal integration in the Mitotic Exit Network occurs through the regulation of multiple signaling elements rather than a central signaling hub. Regulation of the GTPase by spindle position and regulation of the downstream kinase cascade by CDK activity creates a logical 'AND gate'. MEN activation only occurs when (1) a SPB enters the bud and (2) mitotic CDK activity has decreased. We propose that regulating the MEN signaling cascade in series decreases susceptibility to noise; a downstream dependence on CDK inactivation insulates the MEN against spindle movement prior to anaphase. Our work also provides an explanation for why signaling pathways are multi-step processes. Signal integration rather than signal amplification may be the reason for why signal transduction pathways are multi-step processes.

### CDK regulation of the MEN creates anaphase specific ultrasensitivity

Positive feedback loops, such as the mutual regulation between Cdc14 and the MEN (*Figure 7A,B*) as well as double-negative feedback loops, such as exist between CDKs and the MEN (*Figure 7A, C*), create ultrasensitive, switch-like responses (reviewed in *Ferrell and Ha, 2014a*; *Ferrell and Ha, 2014b*; *Ferrell and Ha, 2014c*). In MEN regulation this ultrasensitivity is isolated to anaphase because of its dependence on APC/C-Cdc20 activity. APC/C-Cdc20 activates Cdc14 through the FEAR network and decreases mitotic CDK activity through mitotic cyclin degradation. This primes the MEN for robust, anaphase specific signaling.

Observations of MEN activation prior to anaphase clarify the importance of the feedback loops governing MEN activity. When a SPB moves into the bud during metaphase a slight, but appreciable MEN activation occurs. Small amounts of Mob1 localize to the SPBs, yet signaling strength (as judged by Mob1 association with SPBs) is weak and inconsistent, never leading to release of Cdc14 from the nucleolus. This emphasizes the importance of the FEAR network and mitotic cyclin degradation at the metaphase – anaphase transition. Without the FEAR network, the Cdc14 positive feedback loop is disrupted, slowing MEN activation by 20 min. Likewise, without APC/C-Cdc20 degradation of Clb2 and Clb5 at the metaphase – anaphase transition, the double-negative feedback loop between mitotic CDKs and the MEN is too heavily weighted towards CDK activity, preventing MEN activation.

### Signal integration across multiple serial MEN signaling elements

The initial characterization of MEN regulation focused on the MEN GTPase Tem1. Tem1 was found to make the MEN responsive to spindle position and DNA damage (*Chan and Amon, 2010*; *Hu et al., 2001*; *Cooper and Nelson, 2006*; *Geymonat et al., 2009*). Our work has revealed that MEN regulatory signals control not just Tem1 but also every subsequent step in the pathway (*Figure 7A*). The protein kinase Cdc15 is regulated by the polo-like kinase Cdc5 (*Rock and Amon, 2011*) and CDK activity (*Jaspersen and Morgan, 2000*; *Xu et al., 2000*; this study). The Mob1-Dbf2 kinase is regulated by CDK activity (*König et al., 2010*; this study). These observations raise an important question. Why is the MEN regulated via a serial process rather than integrating all signals through the GTPase?

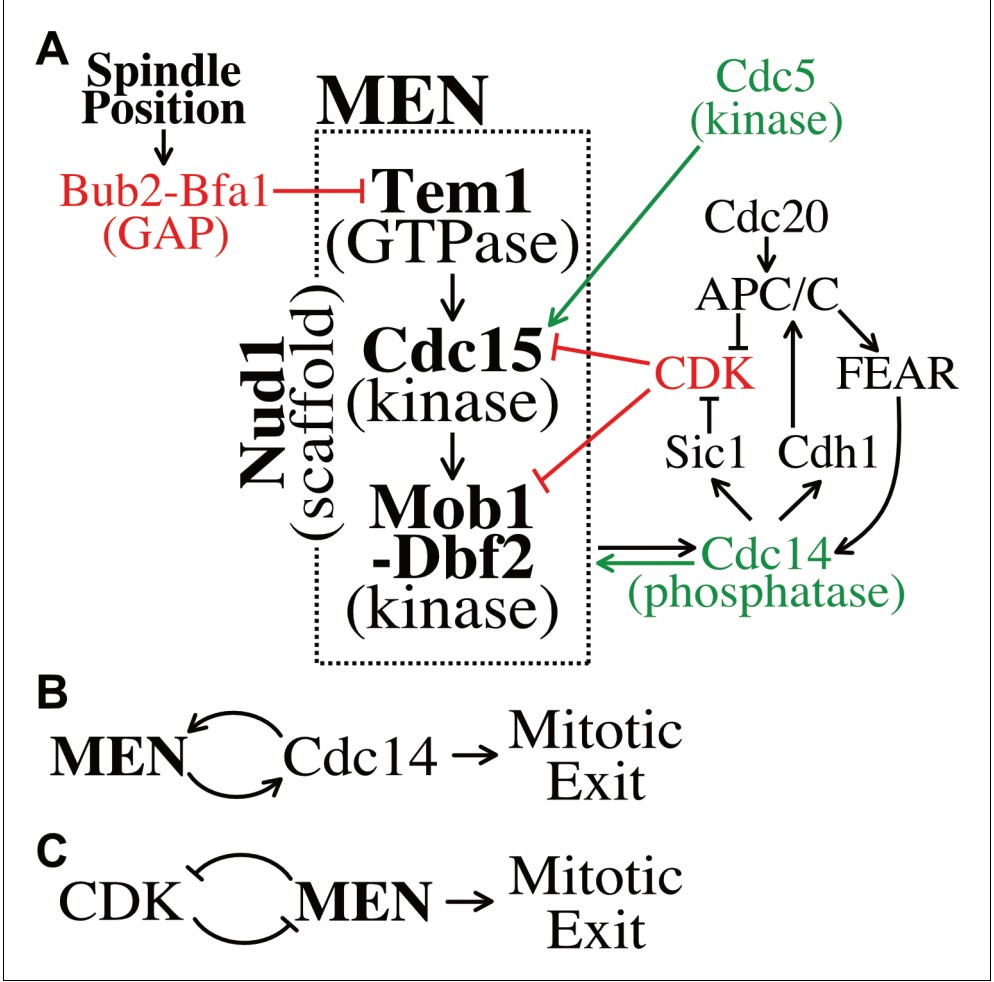

**Figure 7.** Temporal and spatial signals distribute regulation across multiple MEN components. (**A**) The Mitotic Exit Network and its regulators. Direct positive regulators are shown in green, direct negative regulators in red. See text for details. (**B**) The MEN and Cdc14 act in a positive feedback loop to make mitotic exit ultrasensitive. (**C**) CDK and the MEN regulate each other through a double-negative feedback loop isolating mitotic exit ultrasensitivity to anaphase.

DOI: https://doi.org/10.7554/eLife.41139.014

The phenotype of the *cdc15-7A mob1-2A* double mutant indicates that control of the MEN kinases protects cells from spindle movements into the bud such as occur when cells arrest in metaphase for prolonged periods of time. It is worth emphasizing that movement of the nucleus into the bud during metaphase arrests is not merely a byproduct of cell cycle arrests brought about by mutations in genes critical for anaphase entry. DNA damage or microtubule – kinetochore attachment defects can arrest cells in metaphase for hours. Indeed, our live cell analyses show that wild-type cells arrest in metaphase at an appreciable frequency presumably due to light-induced DNA damage. In 67% of these arrested cells the spindle moves in the bud. Our data further suggest that even transient excursions of the spindle into the bud during metaphase could lead to inappropriate Tem1 activation. 15% of *cdc15-7A mob1-2A* double mutants exited mitosis during the metaphase arrest without obvious and hence inferred transient movement of the nucleus into the bud. However, we cannot exclude the possibility that the mitotic exit observed in these 15% of cells is due to partial loss of spindle position sensing. In either case our findings indicate that inhibition of the MEN kinase cascade by mitotic CDKs prevents inappropriate mitotic exit prior to anaphase.

Although the phenotype of the *cdc15-7A mob1-2A* double mutant was similar to that of the *clb2-VI* mutant, it is possible that mitotic CDKs regulate additional MEN components. Mass spectrometry

studies have detected phosphorylation consistent with modification by CDKs on many MEN components and regulators including Bfa1, Cdc5, Cdc14, Lte1, Nud1, Tem1, and Dbf2 (*Ubersax et al., 2003*; *Holt et al., 2009*; *Jones et al., 2011*). The MEN GTPase Tem1 harbors five putative, minimal CDK consensus motifs (S/T-P), but mutation of these sites caused partial loss of Tem1 function (I.W. C. unpublished observations). Other MEN components could however be inhibited by CDKs.

## Serial signal integration beyond the Mitotic Exit Network

The MEN is a member of the Hippo pathway family (Reviewed in *Avruch et al., 2012*; *Hergovich and Hemmings, 2012*). While the core components of the Hippo pathway are highly conserved across eukaryotes, the signaling cascade appears to be fast evolving as judged by the differences in activating signals and targets across species. In budding and fission yeast, the Hippo pathway regulates cytokinesis (*Wolfe and Gould, 2005*), in animals it controls cell growth (*Halder and Johnson, 2011*). Is serial signal integration a common feature of Hippo signaling despite the fast-evolving nature of the pathway? In the fission yeast MEN equivalent, the Septation Initiation Network (SIN), this may be the case. Like in budding yeast the SIN GTPase Spg1 responds to spindle position (*García-Cortés and McCollum, 2009*). In addition, the SIN is inhibited by CDK activity (*Dischinger et al., 2008*; *Guertin et al., 2000*). Likewise, the mammalian Hippo pathway responds to multiple signals including cell polarity, GPCR signaling, and mechanical signals from the extracellular matrix (reviewed in *Yu and Guan, 2013*). The MEN offers a paradigm for how signal integration could occur in mammalian Hippo signaling.

# Materials and methods

## Yeast strains and growth conditions

All *S. cerevisiae* strains in this study are derivatives of W303 (A2587) and are listed in *Supplementary file 1*. Growth conditions are described in the figure legends. Strains were constructed and manipulated as previously described in *Guthrie and Fink (1991)*.

## Plasmid construction

Plasmids used in this study are listed in *Supplementary file 2*.

## Live cell microscopy

The growth conditions for live cell imaging are described in the figure legends. Cells were imaged on agarose pads (2% agarose, synthetic complete (SC) medium containing 2% glucose, unless otherwise noted) affixed to a glass slide and covered with a coverslip. The microscope platform was a DeltaVision Elite (GE Healthcare Bio-Sciences, Pittsburgh, PA) consisting of an InsightSSI solid state light source, an UltimateFocus hardware autofocus system and a model IX-71, Olympus microscope controlled by SoftWoRx software. A 60X Plan APO 1.42NA objective and CoolSNAP HQ2 camera were used for image acquisition. 6 $z$ sections 1 µm apart were acquired and maximally projected.

Image processing was performed using Volocity (PerkinElmer) software. Twenty non-mitotic cells that subsequently move at least one SPB into the bud (*Figure 1C–E*), reached anaphase (*Figure 3B–D*; 4B), or exited mitosis (*Figure 5E,F*, except *cdc20-1* which never exit mitosis; *Figure 6D*) were randomly chosen for analysis. Length of analysis is described in the figure legends. Mob1 intensity was quantified by manually segmenting the SPBs, as identified by Spc42-mCherry localization, and measuring maximum Mob1 intensity within the segmented region [max(SPB)]. Cellular background was subtracted from the maximum intensity at the SPB [max(SPB)-mean(cell)]. Cdc14 release from the nucleolus and NLS$_{Cdc14}$ translocation into the cytoplasm was quantified by manually segmenting the cell and measuring the coefficient of variation (CV = stdev/mean) of intensities within the cell as previously described (*Neurohr and Mendoza, 2017*), normalized and centered as indicated in the figure legends.

## Nud1pT78 immunoblot

Nud1-3V5 immunoprecipitation, Nud1pT78 detection, and Nud1-3V5 detection were performed as previously described (*Rock et al., 2013*) with the following modifications: Denaturing immunoprecipitations were performed on approximately 15 ODs of cells using 30 µl of anti-V5 agarose affinity

gel (Sigma). Primary antibodies: Nud1pT78 was detected with a polyclonal antibody (rabbit 41040, 1:1000 dilution) (*Rock et al., 2013*), Nud1-3V5 was detected with a monoclonal anti-V5 antibody (Invitrogen, 1:2000 dilution).

## Dbf2 kinase assays

Dbf2 kinase assays were performed as previously described (*Visintin and Amon, 2001*) with the following modifications: Immunoprecipitations were performed with an anti-Myc agarose affinity gel (Sigma). Kinase reactions were incubated for 1 hr with gentle mixing and separated by SDS PAGE. Histone H1 phosphorylation was quantified using the phosphorImaging system. Prior to this analysis, the top part of the gel was cut off the kinase assay gel and processed for Dbf2 western blot analysis. Proteins were transferred onto nitrocellulose using semi-dry blotting. The membrane was probed with an anti-c-Myc [9E10] antibody (abcam, 1:500 dilution) and imaged using the ECL Plus system (GE Healthcare).

## Fixed cell microscopy

Indirect in situ immunofluorescence microscopy to detect Tub1 was performed as previously described (*Kilmartin and Adams, 1984*) using a rat anti-α-tubulin (YOL1/34, Oxford Biotechnology, 1:100 dilution). An anti-rat Cy3 antibody (Jackson Laboratory) was used as a secondary antibody. Cells were scored on a Zeiss Axio Observer Z1 inverted microscope (Zeiss. Thornwood, NY).$\geq$100 cells were counted per time point.

## Antibody information

Primary antibody and dilution are listed under individual experimental procedures. Secondary antimouse and anti-rabbit antibodies were used at a 1:5000 dilution (GE).

## Acknowledgements

We thank D Pellman (Harvard, USA), F Uhlmann (LRI, UK), D Morgan (UCSF, USA), M Tyers (UdeM, CA), S.Lacefield (IU, USA), E Schiebel (ZMBH, Germany), and R Deshaies (Caltech, USA) for strains and reagents; members of our laboratory for critical reading of the manuscript. This work was supported by a grant from the Eunice Kennedy Shriver National Institute of Child Health and Human Development (HD085866). AA is also investigator of the Howard Hughes Medical Institute and the Paul F Glenn Center for Biology of Aging Research at MIT.

## Additional information

### Funding

| Funder | Grant reference number | Author |
|---|---|---|
| Eunice Kennedy Shriver National Institute of Child Health and Human Development | HD085866 | Angelika Amon |
| Howard Hughes Medical Institute | | Angelika Amon |

The funders had no role in study design, data collection and interpretation, or the decision to submit the work for publication.

### Author contributions

Ian Winsten Campbell, Conceptualization, Formal analysis, Investigation, Methodology, Writing—original draft; Xiaoxue Zhou, Formal analysis, Investigation, Methodology, Writing—review and editing; Angelika Amon, Conceptualization, Supervision, Funding acquisition, Writing—original draft, Project administration

## Author ORCIDs
Ian Winsten Campbell (iD) http://orcid.org/0000-0003-3019-2560
Xiaoxue Zhou (iD) http://orcid.org/0000-0002-4551-0608
Angelika Amon (iD) http://orcid.org/0000-0001-9837-0314

## Decision letter and Author response
Decision letter https://doi.org/10.7554/eLife.41139.019
Author response https://doi.org/10.7554/eLife.41139.020

## Additional files

### Supplementary files
• Supplementary file 1. Yeast strains. Yeast strains used in this study.
DOI: https://doi.org/10.7554/eLife.41139.015

• Supplementary file 2. Plasmids. Plasmids used in this study.
DOI: https://doi.org/10.7554/eLife.41139.016

• Transparent reporting form
DOI: https://doi.org/10.7554/eLife.41139.017

### Data availability
All data generated or analysed during this study are included in the manuscript and supporting files.

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
