## [Decision Letter]

Thank you for submitting your article "The Mitotic Exit Network integrates temporal and spatial signals by distributing regulation across multiple components" for consideration by *eLife*. Your article has been reviewed by three peer reviewers, and the evaluation has been overseen by a Reviewing Editor and Anna Akhmanova as the Senior Editor. The following individual involved in review of your submission has agreed to reveal his identity: Yves Barral (Reviewer #1).

The reviewers have discussed the reviews with one another and the Reviewing Editor has drafted this decision to help you prepare a revised submission.

Summary:

Campbell et al. investigate the nature of the signals that trigger mitotic exit in yeast. Using a large panel of imaging and biochemical readouts, they show that mitotic exit requires the convergence of two independent signals, (1) the proper entry of a spindle pole into the bud and (2) the activation of APC, which lowers Cdk1 activity. Their conclusion is that all stages of the MEN are responsive to regulation with a spatial control on Bfa/Bub2 and CDK control of Cdc15 and Dbf2 combining to function as an AND gate to provide greater fidelity of MEN activation.

Essential revisions:

Three major issues should be addressed in the revision. The first is that the studies on DNA damage should be extended since currently only the case of *cdc13* mutant cells is considered, which might be a very specific case. The data should be confirmed with other methods for damaging the DNA and more rigorous analyses, for example: on Pds1 degradation in the different mutants.

The second difficulty of the paper is that it uses a flurry of different assays, many of which are used for the first time without describing and validating them thoroughly, and without using them consistently throughout the paper. For example, the assay with Cdc14-NLS is novel, is used only for late parts of the paper, and is not truly validated. In contrast, more direct and better established assays for mitotic exit, such as actomyosin ring contraction, are not used and it is unclear why. More consistent assays should be used throughout the paper.

The third issue is that the paper requires substantial rewriting since it confuses mitotic exit and the mitotic exit network. The only readout of the network that the authors consider is the late one, namely mitotic exit. Early readouts that would be highly relevant to monitor the activation of the MEN, such as Hmt1 phosphorylation (Messier et al., 2013), Kar9 localization, SPB differentiation and the positioning of the metaphase spindle (Hotz et al., 2012a and 2012b; Scarfone et al., 2015; Lengefeld et al., 2017), phosphorylation of the kinesin Kip2 (Drechsler et al., 2015) are omitted. Therefore, many conclusions about the activation of the MEN are not assayed directly, only the activation of mitotic exit by the MEN. In this respect, the published literature concerning the early functions of the MEN should be properly cited; indeed some of the findings presented in the paper have been predicted by others (e.g.: Hotz and Barral, 2014).

---

## [Author Response]

Essential revisions:Three major issues should be addressed in the revision. The first is that the studies on DNA damage should be extended since currently only the case of cdc13 mutant cells is considered, which might be a very specific case. The data should be confirmed with other methods for damaging the DNA and more rigorous analyses, e.g.: on Pds1 degradation in the different mutants.

The study of DNA damage has been expanded to include a second mutant, *cdc9-1* (DNA ligase) and hydroxyurea (nucleotide depletion). We obtained the same results as with the *cdc13-1* mutant. These new data are shown in Figure 5G. Examining Pds1 levels in this experiment does not provide any additional information. Cells do not exit from mitosis when lowering CDK activity but remain arrested in metaphase. Hence, Pds1 degradation has not occurred.

The second difficulty of the paper is that it uses a flurry of different assays, many of which are used for the first time without describing and validating them thoroughly, and without using them consistently throughout the paper. For example, the assay with Cdc14-NLS is novel, is used only for late parts of the paper, and is not truly validated. In contrast, more direct and better established assays for mitotic exit, such as actomyosin ring contraction, are not used and it is unclear why. More consistent assays should be used throughout the paper.

We believe that the strength of our manuscript is that we assay both MEN activity *and* mitotic exit. We would like to point out that we observe a remarkable consistency between assays for MEN activity (Nud1-T78 phosphorylation, Mob1 association with SPBs, Dbf2 kinase activity, Cdc14 release from the nucleolus, Cdc14_NLS_ export into the cytoplasm) and procedures measuring exit from mitosis (anaphase spindle disassembly, loss of Cdc3 from the bud neck, formation of a new bud). However, we acknowledge that this can make the manuscript difficult to follow. Thus, we have edited the text and figures to draw attention to how the various measures of MEN activity correlate with measures of mitotic exit. Every biochemical assay is accompanied by tubulin immuno-fluorescence to assess anaphase spindle disassembly and hence exit from mitosis. To the analysis of Mob1 and Cdc14 localization we have added analyses assessing the formation of new buds, which is yet another assay to assess exit from mitosis. Thus, every experiment provides a measure of MEN activity and exit from mitosis.

We developed the Cdc14_NLS_ nuclear export assay because we required an assay to measure both mitotic exit and MEN activation in the same cell.This was not possible with existing techniques. We would like to emphasize that we did validate the Cdc14_NLS_ assay, demonstrating that it is responsive to modulating MEN but not FEAR network activity. These validations are shown in Figure 5—figure supplement 1. We expanded the description of these validations in the text.

The third issue is that the paper requires substantial rewriting since it confuses mitotic exit and the mitotic exit network. The only readout of the network that the authors consider is the late one, namely mitotic exit. Early readouts that would be highly relevant to monitor the activation of the MEN, such as Hmt1 phosphorylation (Messier et al., 2013), Kar9 localization, SPB differentiation and the positioning of the metaphase spindle (Hotz et al., 2012a and 2012b; Scarfone et al., 2015; Lengefeld et al., 2017), phosphorylation of the kinesin Kip2 (Drechsler et al., 2015) are omitted. Therefore, many conclusions about the activation of the MEN are not assayed directly, only the activation of mitotic exit by the MEN. In this respect, the published literature concerning the early functions of the MEN should be properly cited; indeed some of the findings presented in the paper have been predicted by others (e.g.: Hotz and Barral, 2014).

The Barral lab previously reported that low levels of MEN activity control Kar9 localization during metaphase and spindle pole inheritance. In our manuscript, we specifically analyzed MEN’s anaphase function that promotes Cdc14 release from the nucleolus and exit from mitosis. As requested by the reviewers we have now analyzed Kar9 localization and spindle pole inheritance in MEN mutants. This analysis is shown in new Figure 1—figure supplement 1. Unfortunately, we do not observe this previously reported metaphase function of the MEN. Neither Kar9 localization nor SPB inheritance was altered in cells lacking MEN function. We cannot explain this discrepancy besides to point to a difference in strain background. Given that we have no evidence for MEN activity during metaphase, we have focused our analyses on the well-established role of the MEN in anaphase. All this is described in a new section entitled “MEN activity is confined to anaphase”. The data are presented in new Figure 1—figure supplement 1.